# Kiwifruit Cross-Pollination Analysis: Characterisation of the Pollinator-Assemblage and Practices to Enhance Fruit Quality

**DOI:** 10.3390/plants14162580

**Published:** 2025-08-20

**Authors:** Facundo René Meroi Arcerito, Mariana Paola Mazzei, Camila Corti, María Belén Lezcano, Gregorio Fernández de Landa, Mateo Fernández de Landa, Azucena Elizabeth Iglesias, Facundo Ramos, Natalia Jorgelina Fernández, Natalia Damiani, Liesel Brenda Gende, Darío Pablo Porrini, Matias Daniel Maggi, Leonardo Galetto

**Affiliations:** 1Instituto de Investigaciones en Producción Sanidad y Ambiente (IIPROSAM), CONICET-UNMdP, Centro de Asociación Simple CIC PBA, Mar del Plata 7600, Buenos Aires, Argentina; facundomeroiarcerito@gmail.com (F.R.M.A.); camila.corti20@gmail.com (C.C.); lezcanobele@gmail.com (M.B.L.); gregoriofdl@gmail.com (G.F.d.L.); azucenaelizabeth7@gmail.com (A.E.I.); facun2ramos@gmail.com (F.R.); njfernandez84@gmail.com (N.J.F.); nataliadamiani@gmail.com (N.D.); lieselgende@gmail.com (L.B.G.); mmaggibio@gmail.com (M.D.M.); 2Centro de Investigaciones en Abejas Sociales, Facultad de Ciencias Exactas y Naturales, Universidad Nacional de Mar del Plata, Deán Funes 3350, Mar del Plata 7600, Buenos Aires, Argentina; mateofndl@gmail.com; 3Instituto de Investigaciones en Ciencias Agrarias de Rosario, (IICAR-CONICET), Facultad de Ciencias Agrarias de Rosario, Universidad Nacional de Rosario, Zavalla 2123, Santa Fe, Argentina; marianapmazzei@gmail.com; 4Museo de Ciencias Naturales “Lorenzo Scaglia”, Av. Libertad 3099, Mar del Plata 7600, Buenos Aires, Argentina; dporrini@gmail.com; 5Instituto Multidisciplinario de Biología Vegetal (IMBIV, CONICET-UNC), Universidad Nacional de Córdoba (UNC), Córdoba 5000, Córdoba, Argentina

**Keywords:** *Actinidia deliciosa*, dioecious flowers, honeybees, pollination service

## Abstract

Kiwifruit (*Actinidia deliciosa*) is a globally important crop presenting challenges for ensuring cross-pollination. This study aimed to (1) record the entomological fauna visiting flowers; (2) evaluate the visitation frequency of pollinators; and (3) test the use of lavender extract to enhance cross-pollination by honeybees and assess the impacts on fruit quality. Nine species of floral visitors were recorded as pollinators, although the most frequent were the exotic honeybee (*Apis mellifera*) and the native bees *Bombus pauloensis* and *Xylocopa augusti*. Honeybees increased their visitation to flowers when the attractant was used, improving pollination service and fruit quality compared to the control-bagged treatment, resulting in fruits that were 20 g heavier (115.4 g vs. 95.6 g, 95% CI). Similarly, the number of seeds per fruit and the fruit shape index (FSI) increased in treatments exposed to bee visitation when compared to the bagged control. However, differences in bee visitation among treatments suggested a non-linear relationship between bee activity and fruit quality. Nevertheless, achieving high-quality fruit standards across treatments could be explained by the extended floral lifespan, which allowed for a high number of visits and ensured pollination. Finally, we did not observe any bias in honeybee visitation by applying sugar syrup combined with the attractant. Hence, to increase honeybees’ visits to flowers, we recommend applying the scent directly in a water solution.

## 1. Introduction

Kiwifruit (*Actinidia deliciosa*) is a globally important crop with significant development in Argentina [1], becoming one of the leading producers, although the crop still faces challenges in achieving high-quality standards. The dioecious nature of *A. deliciosa* requires cross-pollination, which can occur via both entomophilous and anemophilous pollination. However, the absence of floral nectar limits the effectiveness of some insect pollinators as pollen vectors [2,3,4,5,6,7]. As a result, considerable efforts have been made to improve kiwifruit production by enhancing insect-mediated pollination as a key strategy [8]. Although *Apis mellifera* can pollinate flowers of kiwifruit [9,10], other bees, such as bumblebees and carpenter bees, may serve as more efficient pollinators because they can extract pollen more effectively by buzzing the anthers—a mechanism known as buzz-pollination [11,12,13]. Pollination efficiency among pollinator species depends on their abundance, morphological traits, and behaviour when collecting pollen (e.g., body size, use of buzzing, time spent on each flower, and visitation frequency) [14].

*Apis mellifera* colonies are generally introduced within the kiwifruit stand to provide pollination services, and the placement of the hives appears to be relevant in the orchard (e.g., number of colonies per hectare and how they are distributed) [15,16,17,18,19]. *Apis mellifera* forages on kiwifruit pollen even when surrounded by other nectar-producing flowers [16]. Among other floral traits, *A. mellifera* can potentially distinguish between flower types by their scent [20]. However, other authors have claimed that *A. mellifera* cannot assess the nutritional value of pollen types based on olfactory cues [21].

Worldwide, kiwifruit growers seek to optimise production according to commercial standards, identifying the key variables for fruit quality, such as weight, number of seeds, and shape [22,23]. Fruit quality can be linked to the degree of pollination, where more deposition of fertile pollen on stigmas leads to a higher number of seeds, which is positively correlated to fruit weight (e.g., [17,22,23,24]).

In Argentina, kiwifruit pollination is primarily managed through human, manual pollination and the introduction of honeybee colonies, with lesser reliance on wind or native entomophilous pollination [19,25]. Nevertheless, the specific impacts on fruit quality of pollination by honeybees remain to be experimentally evaluated. Therefore, field trials measuring the impact of entomophilous pollination on crop production are needed, taking into account variations in the behaviour and contribution of different bee species, and analysing technologies aimed to enhance cross-pollination, such as the use of attractants.

Previous studies have investigated the use of floral odours to enhance pollinator perception, learning, and behaviour following exposure inside the hive [26,27]. Additionally, Meroi Arcerito et al. (2021) showed that lavender extract was an effective attractant for honeybees, enhancing their visits to female flowers, but tested only on female plants [23]. Therefore, we aimed to expand upon the knowledge gained from this preliminary study through the following objectives: 1. Record, for the first time in Argentina, the entomological fauna visiting kiwifruit flowers in a productive matrix within an agroecosystem; 2. Evaluate the visitation frequency of different pollinators to male and female flowers under an artificial intensification scenario of entomological pollination; 3. Test a lavender extract applied to flowers to enhance cross-pollination, evaluating the behaviour of honeybees and the impacts on fruit quality. Hence, we formulated two hypotheses in line with our objectives: (1) a diversity of pollinator taxa is present despite the surrounding agricultural matrix of intensification in other crops but the frequency of visits by *Apis mellifera* to flowers of kiwifruit is expected to be the highest due to the introduction of beehives; (2) an increase in honeybee visitation is expected when adding a floral attractant improving fruit quality. The impacts of increased flower visitation of honeybees on fruit quality were analysed by comparing experimental treatments with and without a floral attractant. This work aimed to develop a new effective pollination strategy, enhancing honeybee visitation and fruit quality.

## 2. Materials and Methods

### 2.1. Area of Study

The study was conducted near Mar del Plata City (Buenos Aires Province, Argentina) on the Santa Paula farm in November 2019 (spring in the southern hemisphere). The climate in this region of Argentina is mostly humid and cool, with strong seasonal variation [23], and no rainfall, storms, or windy days were recorded during the sampling within the flowering season of kiwifruit (National Meteorological Service). *Actinidia deliciosa* “Hayward” is cultivated in soils characterised by thick horizons, rich in organic matter and macronutrients (no need for external inputs such as fertilizers), obtaining acceptable commercial quality standards concerning international markets (personal communication). The kiwifruit vines at Santa Paula were arranged in rows through a pergola system, forming a horizontal canopy at 2 m height with 3 m between plants within each row, and 4 m between rows under a regular pruning system according to commercial standards. The plantation was mainly surrounded by other crops (e.g., pumpkins, hops, and *Eucalyptus* spp.), in an area undergoing regular land-use intensification [27,28]. Farmers typically avoid using chemical agents to promote kiwifruit blooming, and they usually introduce *A. mellifera* hives during flowering and perform manual pollination as a standard practice to ensure fruit production.

### 2.2. Bee Pollination Service

Ten *A. mellifera* beehives per hectare were moved to the plot about seven days before peak kiwifruit blooming (i.e., when fewer than 10% of the flowers had opened). Hives were placed 5 m apart from each other, 2–5 m from the edge of the kiwifruit plantation, and were kept in the crop throughout the entire blooming period (approximately two weeks). The hives were standardised in bee population, brood, and food reserves, and stimulated with sugar syrup [23,28].

*Bombus pauloensis* and *Xylocopa augusti* were present in the crop during blooming. However, we increased the number of *B. pauloensis* by placing five commercial hives at the edge of the half-hectare plot, where experiments and observations of pollinators were performed. The bumblebee colonies were provided by a national supplier (Brometan SRL, Burzaco, Argentina). Both non-*Apis* bees are buzz-pollinators and differ from *A. mellifera* in their daily activity periods and tolerance to some climatic conditions. For example, bumblebees are often active even during the colder parts of the day, when *A. mellifera* typically is not.

### 2.3. Floral Visitors and Pollinators Sampling

The species classified as floral visitors included those that contact only the petals, whereas species classified as pollinators made contact with the sexual organs of both male and female flowers. The observations were conducted on 408 flowers of *Actinidia deliciosa* “Hayward” (173 males and 235 females), separately for male and female individuals. Although male flowers began blooming approximately one week before female flowers, sampling of bee visitation frequency was conducted when at least 75% of the individuals were in bloom. All flowers used in the trials were marked and remained accessible to pollinators, with no hand-pollination performed (hand-pollination, a common practice among growers, involves applying a pollen solution to open flowers). Lavender extracts (Lavandin Grosso from *Lavandula hybrida* hydroalcoholic extract [23]) were applied through a handheld sprayer on several flowers in five female plants and three male plants per treatment. The variable “frequency of visits” was standardised on a per-flower basis (number of visits during 5 min/number of observed flowers) for each data point [23]. We measured the frequency of visits per flower and the time spent per visit on the same flower (seconds, i.e., the contact period between the bees and the flower’s sexual organs, as an indicator of the behaviour of the most frequent pollinator) of each bee-pollinator. To record visits by *Xylocopa augusti* and *Bombus pauloensis*, we monitored one-quarter of the canopy of a given plant. Observations were conducted for 5 min periods twice a day, following the same schedule as *A. mellifera*, and the number of visits was also standardised by the number of flowers followed (number of visits during 5 min/number of observed flowers). However, since *Apis mellifera* visited flowers more frequently than the other bees, sampling for this species was conducted on a smaller number of flowers (approximately 5–9) for 5 min periods per tree; thus, the trial involves about 60 flowers per treatment, considering all the trees together. The frequency of visits was standardised as for the other bees (number of visits during 5 min/number of observed flowers). Summarising, data on the frequency of visits for the different pollinators to male and female flowers of kiwifruit were obtained between 10 a.m. and 2 p.m., for 5 min twice daily (each sampling was considered independent), during the five days corresponding to the blooming peak for this crop in the study site [23].

### 2.4. Impact of the Attractant on Honeybee Pollination, Fruit Production, and Quality

In a pergola system with a ratio of 8 female plants per 1 male plant (8:1), one male individual was planted after every eight female plants within each row. To maximize coverage, the position of male plants was staggered across adjacent rows, so that male plants in one row were offset from those in neighboring rows. This ensured a uniform distribution of male plants throughout the orchard and reduced the distance between female and male plants, thereby increasing pollination efficiency.

Plant and flower selection: five female plants were randomly selected for each treatment to avoid any bias among treatments due to scent proximity (20 female plants in total), excluding border plants and those located immediately adjacent to male plants, thus avoiding border and male proximity effects. To minimise individual plant bias, we treated it as a random factor. We selected seven flowers for each treatment per plant, except for controls, where control—“open pollinated” and control—“bagged” flowers were randomly selected, with one to two flowers per plant involving those plants used for the other treatments. Flowers were always exposed to pollination (except control—“bagged” flowers) and received the treatment 30 to 15 min before the visit census to recreate a possible protocol for producers. The experiment involved four treatments and two types of controls (n = 35 flowers per treatment and each control), identifying the treated flowers with different coloured tapes. To minimise bias in resource allocation, only flowers located in the central section of the main branches were used for the experiment; within this section, flowers were randomly assigned to treatments and marked accordingly before anthesis. This sampling strategy ensured that plant and flower selection within the orchard was both random and representative.

Treatments that required spraying on the flowers were applied from a distance of 10 cm, avoiding damage in floral whorls. The two controls and the treatments were as follows: (a) control-“bagged” flowers (with voile) not sprayed nor insect-pollinated, where only wind could act as pollen vector (control treatment for insect pollinators and the treatments with extract); (b) control-“open pollinated” flowers not sprayed at all (control for exposed treatments to pollinators); (c) “water”, flowers exposed to pollinators and only sprayed once with distilled water (about 0.3 mL per flower) (negative control for the water component of the treatments with extract); (d) “syrup”, exposed flowers to pollinators and sprayed only with a sugar solution (made by a solution 1:1 of commercial sugar diluted in the equal volumetric amount of distilled water; as a negative control of the syrup+extract treatment); (e) “syrup+extract”, exposed flowers to pollinators and sprayed with the syrup+ the lavandin extract (1% *v*/*v*, where 990 mL was made of 1:1 water/sugar and 10 mL of *Lavandula hybrida* extract solution); (f) “extract”, exposed flowers to pollinators and sprayed only with the water + lavandin extract (i.e., water + *Lavandin grosso* extract, 1% *v*/*v*, 10 mL of *Lavandula hybrida* extract solution in 990 mL).

When fruits are ready for harvesting according to commercial standards (6 months after blooming and fruit maturation above 6 points in Brix degrees), 6 fruits per plant were collected and measured. Thus, we measured 30 fruits per treatment. Mature fruits were collected in May 2020 (autumn in the southern hemisphere). The response variables considered were fruit mass (g), measured using a trade balance (Morley BA 402), and the number of seeds per fruit (manually counted as an indicator of pollination efficiency). Each fruit was divided into four equal sections, and then all seeds were counted in only one section to finally estimate the total number of seeds per fruit (# of seeds counted × 4). Fruit mass was considered a proxy for fruit quality, and seed number per fruit was a proxy for pollination efficiency. Finally, we measured each fruit’s maximum and minimum width and length using a caliper to obtain a fruit shape parameter (shape coefficient) (*sensu* [29]). A larger shape coefficient indicates a higher fruit quality for the market, whereas a lower one indicates a rounded fruit, usually related to smaller and lower-quality fruits.

### 2.5. Statistical Analyses

The “frequency of visits by *A. mellifera*” (the number of bees per flower within 5 min) and “time per visit” (recorded in seconds) were modelled using generalised linear mixed models (GLM) with Lognormal errors to determine the effects of the treatments and the flower’s sex.

To assess the effects of the attractant on fruit quality, we used GLMs to compare treatments on the dependent variables: fruit mass (Gaussian errors), number of seeds (Negative Binomial errors), and fruit shape index (FSI) (sensu [29]) (Gaussian errors). We used the glmer.nb and glm functions from the lme4 package in R software version 4.4.2 [30] to conduct the GLMs. We used Tukey tests from multcomp [31] to compare means (*p* < 0.001). To represent the estimates of each selected model, we used plots made with the plot (R Core Team, 2021) and the visreg [32] function.

All statistical analyses were performed using R software version 4.4.2 [33]. We used the function glmer from the lme4 package [30] to conduct the GLMM and function fitdist from the fitdistrplus package [34] to fit the best probability distribution of each dependent variable. Also, we chose the likelihood ratio test with the function ANOVA from the stats package [33] to select the best-fitted model. The assumptions of the linear regression with the binomial family were tested using the DHARMa package [35]. To represent the estimates of each selected model, we used plots made with the ggplot2 [36] and visreg [32] packages.

## 3. Results

### 3.1. The Floral Visitors and Their Visitation Patterns

A total of nine species were recorded in contact with the reproductive organs of the flower (suggesting these species can be considered as potential pollinators of kiwifruit) for the first time in the region: *Paromoeocerus barbicornis (Cerambycidae*, *Coleoptera)*, *Harmomonia axyridis (Coccinellidae*, *Coleoptera)*, *Polybia scutellaris (Hymenoptera*, *Vespidae)*, *Allograpta exotica (Syrphidae*, *Diptera)*, *Toxomerus* sp. *(Syrphidae*, *Diptera)*, *Platycheirus* sp. *(Syrphidae*, *Diptera)*. However, bees (*A. mellifera*, *B. pauloensis*, and *X. august*) visited the flowers consistently and could be considered the main pollinators of this crop in the region. The total abundance registered for these bees during the whole sampling period consists of 517 honeybees, 50 bumblebees, and 47 carpenter bees, with a mean frequency per flower of 1.95, 0.17, and 0.18 bees per 5 min observation, respectively. The mean time-period (seconds) per visit for these species was 10.46, 17.30, and 41.98 s, respectively (Figure 1).

### 3.2. Impact of the Extract on Honeybee Pollination and Fruit Quality

The model considering treatments and the sex of the flowers as independent variables was significant (*p* < 0.001) for the frequency of *A. mellifera*. Honeybees foraged preferentially on female flowers in all treatments (Figure 2A), and the treatments with the extract (i.e., treatments “Extract” and “Syrup+Extract”) attracted significantly more bees than the others (Figure 2A). The model with treatment as an independent variable was significant (*p* < 0.001) for the average foraging period of a honeybee on a flower (Figure 2B and Table 1). The period (seconds) that honeybees spent on flowers was highly variable (Figure 2B).

For all variables of fruit quality (weight, # of seeds, and FSI), treatment as an independent variable fitted better (*p* < 0.001) than the model without an independent variable (null model) (Figure 3A,C,E, Table 2). Tukey’s test showed significant differences between treatment levels (Figure 3B,D,F). There were differences in fruit weight, the number of seeds per fruit, and fruit shape (FSI) among the control-“bagged” treatment and all the other treatments exposed to pollinators (Figure 3) (*p* < 0.001). Besides, the treatment control-“open pollinated” (not bagged nor sprayed) also exhibited lighter fruits than the “syrup+extract”, “extract”, and “water” treatments (Figure 3) (*p* ≤ 0.05). The fruits from the control-“bagged” (only anemophilous pollination) treatment weighed about 20 g less than those from the “extract” treatment (95.60 g, 95% CI [87.07, 104.13], and 115.40 g, 95% CI [106.87, 123.93], respectively). Similarly, there was a difference of ~21 g between the control-“bagged” and “syrup+extract” treatments (95.60 g, 95% CI [87.07, 104.13], and 116.40 g, 95% CI [107.87, 124.93], respectively). The fruits from the “syrup” treatment were very similar in weight to those from the control-“open pollinated” treatment, with a slight difference of 6.2 g (101.80 g, 95% CI [93.27, 110.33] and 95.60 g, 95%CI [87.07, 104.13], respectively).

## 4. Discussion

### 4.1. The Floral Visitors and Their Visitation Patterns

Several floral visitors were observed for the first time in this region, and they can be considered pollinators of this crop. However, non-bee potential pollinators are scarce and, thus, perhaps with a minor contribution to kiwifruit pollination. Nevertheless, further exploration is needed to understand the potential role of minor non-bee pollinators in the cross-pollination of kiwifruit if their abundance is enhanced through seminatural habitat conservation or other environmental policies.

Our results indicate that *A. mellifera* was the most abundant visitor and occurred more frequently within the orchard than other bee species. This result could be expected because honeybee hives are regularly placed within kiwifruit stands. The frequency of native bees (*B. pauloensis* and carpenter bee *X. augusti*) in the kiwifruit flowers was lower than honeybees, despite some colonies of *B. pauloensis* being added in the study site. Although *B. pauloensis* differed from the carpenter bee in the time spent on each flower, both are buzzing bees that pollinate kiwifruit flowers. There is no clear agreement regarding which bee pollinator is the best or the most efficient in cross-pollination for kiwifruit. Nayak et al. [37] mentioned that *Bombus haemorrhoidalis* visited more flowers per min than honeybees and spent more time per flower visit. However, colleague from Spain [8] stated that honeybees were more abundant and display a higher frequency of visits than any other insect, but showed that bumblebees were more efficient on a single-visit basis concerning the impact on fruit quality; they also mentioned other taxa that contribute less because of their lower numbers (hoverflies, wild bees, butterflies, beetles) or their passive behaviour (non-syrphid flies). For the Gold Kiwi (*Actinidia chinensis* P. ‘Haegeum’), *Apis mellifera* could be a better pollinator than *Bombus terrestris* due to their higher abundance [37]. Thus, entomological pollination in *A. deliciosa* should be studied at both regional and local scales, as pollinator populations may vary across these scales, to understand their contribution to fruit set and quality. In this study, *Apis mellifera* had a positive impact on increasing cross-pollination, challenging the recommendation of performing additional hand-delivered pollination methods (e.g., [19,23,25]), which is considered irreplaceable [29]. Honeybees foraged preferentially on female flowers in all treatments; however, visitation increased in both female and male flowers by using the floral-odour attractant (Figure 3, treatments with *Lavandin grosso* extract).

### 4.2. Impact of the Extract on Honeybee Pollination and Fruit Quality

The experiment with and without attractants showed better fruit quality for all treatments exposed to pollinators than the control-“bagged” flowers in all the variables assessed (weight, seed number per fruit, and shape index), reinforcing the role of insect pollination over wind pollination. Moreover, *Apis mellifera* visited the flowers sprayed with the *Lavandin grosso* extract (“extract” and “syrup+extract” treatments) more frequently than the flowers of the other treatments, increasing honeybee visitation. However, fruit quality did not increase to the same extent as flower visitation in the “extract” and “syrup+extract” treatments. Although we observed a tendency toward increased fruit quality in treatments with the attractant, the differences were not statistically significant compared to the other treatments exposed to pollinators. The mismatch between visitation rates and fruit quality may be explained by the extended floral lifespan (approximately 5 days), which allows for a high number of visits, potentially ensuring sufficient pollination for all treatments exposed to pollinators, at least during bloom days that were free of storms or strong winds. In summary, despite increased visitation to flowers added with an odour attractant, fruit quality traits were not always statistically significant, possibly due to the extended floral longevity during this particularly favourable season. In addition, the treatment using only water compared to the control (control-“open pollinated”) also tended to favour fruit weight, despite the lower frequency and high variability of bee visits during observation periods (Figure 3A). Kiwifruit shape index (FSI) and weight increased under lavender extract treatments, as was previously reported for this region [23]. We did not register any bias in honeybee visits when adding sugar syrup with the attractant. Therefore, we recommend applying the scent in a water solution alone to attract more bees to the flowers and improve fruit production in *A. deliciosa*.

## 5. Conclusions

This study expands the knowledge of pollinator dynamics in Argentine kiwifruit orchards, highlighting *Apis mellifera* as the dominant floral visitor. Native bees *Bombus pauloensis* and *Xylocopa augusti* showed distinct foraging behaviour, suggesting differences in pollination efficiency. Honeybees improved fruit quality, and lavender extract increased their visitation. These findings support the use of attractants as tools to enhance or support pollination. We recommend considering local conditions and pollinator communities to improve kiwifruit pollination and fruit quality.

## Figures and Tables

**Figure 1 plants-14-02580-f001:**
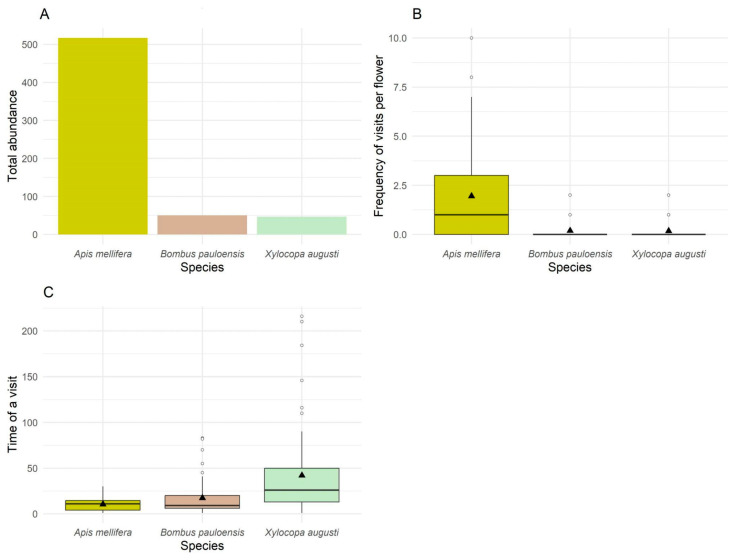
Visitation patterns of the flower visitors: *Apis mellifera, Bombus pauloensis,* and *Xylocopa augustii.* (**A**): Total abundance of each species. (**B**): Frequency of visits per flower (# of visits per flower during 5 min) per species. (**C**): Time of a visit (seconds) per species. In boxplots (**B**,**C**), the mean of each variable is indicated with a black triangle. Circles outside the box indicate outlier values.

**Figure 2 plants-14-02580-f002:**
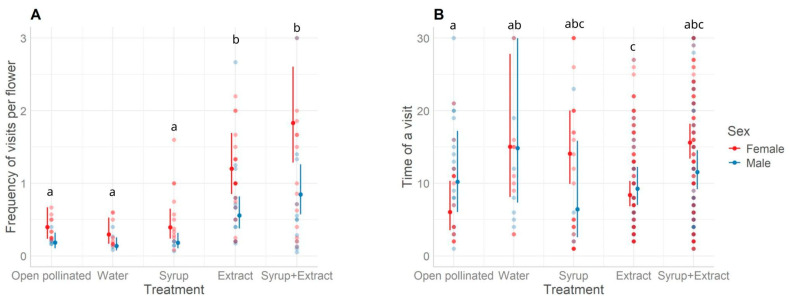
Frequency of visits per flower and time of a visit (seconds) of *Apis mellifera* according to the experimental treatments. (**A**): Frequency of *A. mellifera* visiting male and female flowers per treatment (means and the confidence intervals of 95%). The best-fitted model included the treatment and sex of the flower (Table 1). Female flowers received more visits from honeybees during all treatments (*p* < 0.001). Smallcase letters (a, and b) indicate Tukey contrast (*p* < 0.05) results for each treatment (control-“open pollinated”, “extract”, “syrup”, “syrup+extract”, and “water” as explained in Materials and Methods). (**B**): Duration of visit (pollinating behaviour) of each *A. mellifera* visiting a flower in seconds per treatment (means and the confidence intervals of 95%). The best-fitted model included just treatment. Lowercase letters (a, ab, abc, and c) indicate Tukey contrast (*p* < 0.05) results for each treatment (control-“open pollinated”, “extract”, “syrup”, “syrup+extract”, and “water”). Each dot represents a field measurement.

**Figure 3 plants-14-02580-f003:**
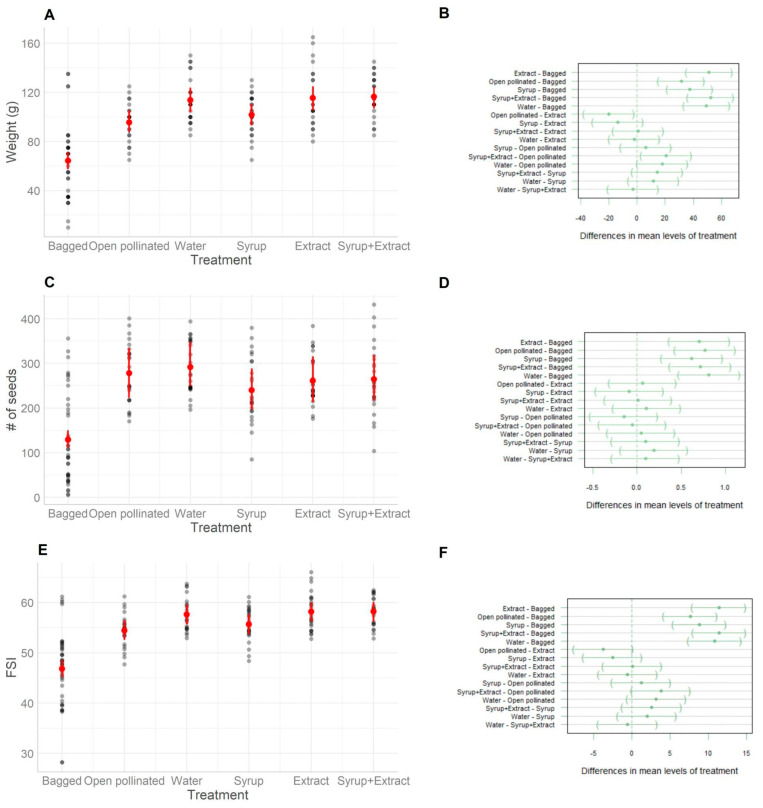
Effects of the experimental treatments with and without attractants on the quality of kiwifruits. (**A**): weight (g) predicted by the model with Gaussian errors (R^2^: 0.48, *p* < 0.001). (**B**): Tukey contrast for fruit weights between each level of the model’s independent variable (treatments). (**C**): # of seeds per fruit (number of seeds of ¼ kiwi) predicted by the model with Negative Binomial errors (R^2^: 0.43, *p* < 0.001). (**D**): Tukey contrasts the number of seeds per fruit between each level of the model’s independent variable (treatments). (**E**): Fruit shape index (FSI) predicted by the model with Gaussian errors (R^2^: 0.49, *p* < 0.001). (**F**): Tukey contrast for FSI between each level of the model’s independent variable (treatments). The control-“bagged” treatment are the flowers restricted in entomophilous pollination, and such process only depended on anemophilous pollination; control-“open pollinated” refers to open flowers pollinated by insects and wind, but these flowers were not bagged nor sprayed; “extract” refers to the extract+water; “syrup” to sugar+water; “syrup+extract” to the extract diluted in water and sugar, and all dilutions as were explained in the materials and methods section; “water” implies only distilled water. In (**A**,**C**,**E**), the plots include the expected mean (red point) and the 95% confidence intervals for the expected value (red band). In (**B**,**D**,**F**), the plots include the results of the differences in means for each treatment and the 95% confidence intervals. Each dot represents a field measurement.

**Table 1 plants-14-02580-t001:** Tested models for the treatments exposed to the visits of *Apis mellifera*. The best-fitted model is highlighted in green with its *p*-value. Independent variables: fsex: sex of flowers, null: null model (without independent variables), and “treatment”: experimental treatments applied to flowers (control-“open pollinated”, “extract-*Lavandin grosso*”, “syrup”, “syrup+extract”, and “water”).

Models Tested for Pollinators	*p*-Value
**Dependent variable: Flower frequency**	
Family Lognormal	
*treatment+fsex*	*0.001*
*treatment*	
*null*	
**Dependent variable: Time of a visit**	
Family Lognormal	
*treatment+fsex*	
*treatment*	*0.001*
*null*	

**Table 2 plants-14-02580-t002:** Tested models for the quality of kiwifruit, comparing the six experimental treatments with *Lavandin grosso* and without attractants. The best-fitted model is indicated in green with its *p*-value. Independent variable: treatment (control-“bagged”, control-“open pollinators”, “extract”, “syrup”, “syrup+extract”, and “water”), null: null model (without independent variables).

Models Tested for Quality of Fruit	*p*-Value
**Dependent variable: Weight (g)**	
Family Gaussian	
*treatment*	*0.001*
null	
**Dependent variable: # of seeds**	
Family Negative Binomial	
*treatment*	*0.001*
null	
**Dependent variable: FSI**	
Family Gaussian	
*treatment*	*0.001*
null	

## Data Availability

The original contributions presented in this study are included in the article/Appendix A. Further inquiries can be directed to the corresponding author.

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
