# Peer review of "Kiwifruit Cross-Pollination Analysis: Characterisation of the Pollinator-Assemblage and Practices to Enhance Fruit Quality"

_plants, 2025, doi:10.3390/plants14162580_

Round 1
Reviewer 1 Report (Previous Reviewer 1)
Comments and Suggestions for Authors
In this manuscript, Facundo René Meroi Arcerito and colleagues recorded the entomological fauna visiting flowers, evaluated the visitation frequency of pollinators; and tested the use of lavender extract to enhance cross-pollination by honeybees, and assess the impacts on fruit quality. I have following comments:
1, For the Abstract section, more values should be presented. For instance, authors stated that “the number of seeds per fruit and the fruit shape index (FSI) increased in treatments exposed to bee visitation when compared to the bagged control.” , but the values were not provided.
2, For the introduction section, practical interests of this study should be stated in the last paragraph.
3, Genotypes of Kiwifruit and pollinators examined in this study should be described in the Materials
4, Methods for random sampling should be introduced in the Methods.
5, For the results, individual data points should be shown in each bar chart in Figures 2 and 3
6, Data described in the subsection 3.1 should be exhibited by Figure.
7, For the conclusion section, authors should consider to combine the two paragraphs into one paragraph.
Author Response
We would like to thank you and the reviewers for your invaluable support and thoughtful comments throughout the editorial process. Such feedback significantly improved the quality of our manuscript, and we are truly grateful for the opportunity to publish our work in this esteemed journal. Below, you will find a point-by-point response to the reviewers' inquiries, marked in blue, along with some recommendations from the editor.
Thank you again for your kind assistance and for considering our contribution.
Warm regards.

Reviewer 2 Report (Previous Reviewer 2)
Comments and Suggestions for Authors
The only comment I have is a follow-up on comment no. 1 from the first version - please add the male cultivar and specify the planting ( I believe 1:8 means a male every third plant in every third row)
Author Response
Dear Reviewer 2,
We would like to thank you and the reviewers for your invaluable support and thoughtful comments throughout the editorial process. Such feedback significantly improved the quality of our manuscript, and we are truly grateful for the opportunity to publish our work in this esteemed journal. Below, you will find a point-by-point response to the reviewers' inquiries, marked in blue, along with some recommendations from the editor.
Thank you again for your kind assistance and for considering our contribution.
Warm regards.

Reviewer 3 Report (Previous Reviewer 3)
Comments and Suggestions for Authors
17 July, 2025
Dear Editor Rojyotin Tiamsai
Manuscript ID: plants-3790231
Title of the manuscript: Kiwifruit Cross-Pollination Analysis: Characterisation of the Pollinator-Assemblage and Practices to Enhance Fruit Quality
In the study, information about “Kiwifruit cross-pollination analysis: Characterisation of the pollinator-assemblage and practices to enhance fruit quality” is provide.
I saw that the suggestions I made during the initial review of the manuscript were implemented to point to point. The MS has been significantly improved with the author's revisions, based on my suggestions and those of other reviewers. The MS contains valuable information, particularly regarding solutions to the pollination-related yield deficits faced by kiwifruit growers. Because it is well-written and contains original findings, I believe the study will be followed with interest by readers. Following these revisions, the MS is now accepted for publication in PLANTS.
Note: Minor suggestions were shown on the annotated PDF file.
With my best regards

Author Response
Dear Reviewer 3,
We would like to thank you and the reviewers for your invaluable support and thoughtful comments throughout the editorial process. Such feedback significantly improved the quality of our manuscript, and we are truly grateful for the opportunity to publish our work in this esteemed journal. Below, you will find a point-by-point response to the reviewers' inquiries, marked in blue, along with some recommendations from the editor.
Thank you again for your kind assistance and for considering our contribution.
Warm regards.

Reviewer 4 Report (New Reviewer)
Comments and Suggestions for Authors
This research is very important and relevant, with a good theoretical fundamentation and the statistical analysis is appropriated. We recommend to the authors review the results and data presented for being better and the discussion more detailed.
However, the conclusion should be a single paragraph, no longer than five lines, and should be linked to the objective. It is too long, with data that should be included in the results and or discussion.
Author Response
Dear Reviewer 4,
We would like to thank you and the reviewers for your invaluable support and thoughtful comments throughout the editorial process. Such feedback significantly improved the quality of our manuscript, and we are truly grateful for the opportunity to publish our work in this esteemed journal. Below, you will find a point-by-point response to the reviewers' inquiries, marked in blue, along with some recommendations from the editor.
Thank you again for your kind assistance and for considering our contribution.
Warm regards.

Round 2
Reviewer 1 Report (Previous Reviewer 1)
Comments and Suggestions for Authors
Authors have addressed my concerns in the revision.
This manuscript is a resubmission of an earlier submission. The following is a list of the peer review reports and author responses from that submission.
Round 1
Reviewer 1 Report
Comments and Suggestions for Authors
In this manuscript, Facundo René Meroi Arcerito and colleagues recorded the entomological fauna visiting kiwifruit flowers for the first time in Argentina, evaluated the visitation frequency of different pollinators to male and female flowers and tested the use of lavender extract to enhance cross-pollination by honeybees, evaluate their behaviour under different experimental treatments, and assess the impacts on fruit quality. I have following comments:
1, For the Title, I suggest to employ “Kiwifruit cross-pollination analysis: characterisation of the pollinator-assemblage and practices to enhance fruit quality”.
2, For the Abstract, detailed data should be presented. For instance, authors stated that “However, fruit quality did not match the extent of the increase in flower visitation by honeybees across all experimental treatments”, and number should be provided.
3, For the key words, honeybee should be included.
4, For the introduction, main conclusions and practical interests of this study should be stated in the last paragraph of this section.
5, The section 3 "Impact of the AĴractant on Honeybee Pollination and Fruit Quality" is confusing, please optimize by incorporating this section into Materials and methods or Results
5, For the Results, error bar should be included in the Figures 1 and 2.
6, For the Discussion section, I would like to see this section could be divided into several subsections and each subsection is properly entitled.
7, For the Materials and methods, genotypes of plant materials employed in this study should be introduced and randomization in the samplings should be clearly described. Location of experiment sites should be indicated in a map
8, A conclusion section should be included.
Reviewer 2 Report
Comments and Suggestions for Authors
- When describing the orchard, you should mention the male/female ratio and male position.
- Counting seeds in a fourth of a fruit can be biased, as underpollinated fruits are asymmetric in seed distribution.
- The number of seeds per fruit should be displayed for all fruits in the graphs. Graphs should be changed to show the average (with plus and minus boxes) instead of all fruits.
- When trying to determine the best insect pollinator, you need a "natural" habitat with no introduced insects, as they change the environment. you missed an important measurement for pollinator quality - pollination after a single visit.
- The only significant differences in seeds per fruit and fruit size (which you don't mention that are correlated) are between the bagged flowers to all other treatments. the meaning is that insect pollination is crucial for kiwifruit pollination, and you were not able to increase it with any described treatments.

Reviewer 3 Report
Comments and Suggestions for Authors
10 June, 2025
Dear Editor Rojyotin Tiamsai
Manuscript ID: plants-3689654
Title of the manuscript: Kiwifruit Cross-Pollination: Characterisation of the Pollinator-Assemblage and Practices to Enhance Fruit Quality
In the study, information about “Kiwifruit Cross-Pollination: Characterisation of the Pollinator-Assemblage and Practices to Enhance Fruit Quality” is provide.
The study contains valuable information. However, it is not a research article that meets the standards of the “Plants” journal. It can be evaluated as a research note or short communication. I have shown my criticisms that I think will contribute to the development of the study in the attached PDF file. Therefore, I regret to state that the study cannot be accepted for publication.
Best regards
Note: My suggestions were shown on the annotated PDF file.
With my best regards
